# Paclitaxel and Caffeine–Taurine, New Colchicine Alternatives for Chromosomes Doubling in Maize Haploid Breeding

**DOI:** 10.3390/ijms241914659

**Published:** 2023-09-28

**Authors:** Saeed Arshad, Mengli Wei, Qurban Ali, Ghulam Mustafa, Zhengqiang Ma, Yuanxin Yan

**Affiliations:** 1State Key Laboratory of Crop Genetics and Germplasm Enhancement, Nanjing Agricultural University, Nanjing 210095, China; maliksaeedawan@hotmail.com (S.A.); 2019201097@njau.edu.cn (M.W.); zqm2@njau.edu.cn (Z.M.); 2Department of Plant Pathology, College of Plant Protection, Nanjing Agricultural University, Nanjing 210095, China; rattarqurban@hotmail.com (Q.A.); 2018201104@njau.edu.cn (G.M.); 3Jiangsu Collaborative Innovation Center for Modern Crop Production, Nanjing 210095, China

**Keywords:** maize haploid, chromosome doubling, paclitaxel, caffeine–taurine, colchicine, anticancer

## Abstract

The doubled haploid (DH) technology is employed worldwide in various crop-breeding programs, especially maize. Still, restoring tassel fertility is measured as one of the major restrictive factors in producing DH lines. Colchicine, nitrous oxide, oryzalin, and amiprophosmethyl are common chromosome-doubling agents that aid in developing viable diploids (2n) from sterile haploids (n). Although colchicine is the most widely used polyploidy-inducing agent, it is highly toxic to mammals and plants. Therefore, there is a dire need to explore natural, non-toxic, or low-toxic cheaper and accessible substitutes with a higher survival and fertility rate. To the best of our knowledge, the advanced usage of human anticancer drugs “Paclitaxel (PTX)” and “Caffeine–Taurine (CAF–T)” for in vivo maize haploids doubling is being disclosed for the first time. These two antimitotic and antimicrotubular agents (PTX and CAF–T) were assessed under various treatment conditions compared to colchicine. As a result, the maximum actual doubling rates (ADR) for PTX versus colchicine in maize haploid seedlings were 42.1% (400 M, 16 h treatment) versus 31.9% (0.5 mM, 24 h treatment), respectively. In addition, the ADR in maize haploid seeds were CAF–T 20.0% (caffeine 2 g/L + taurine 12 g/L, 16 h), PTX 19.9% (100 μM, 24 h treatment), and colchicine 26.0% (2.0 mM, 8 h treatment). Moreover, the morphological and physiological by-effects in haploid plants by PTX were significantly lower than colchicine. Hence, PTX and CAF–T are better alternatives than the widely used traditional colchicine to improve chromosome-doubling in maize crop.

## 1. Introduction

During the last century, the development of maize inbred lines relied on recurrent selfing and selection to reach the desired level of homozygosity [1]. It requires six-to-eight generations of selfing after crossing two parent lines. In contrast, the process of double haploid (DH) line production involves generating haploids (by crossing female source parent with an inducer) with only the gametic chromosomes, the identification of haploids from diploids, and the doubling of the chromosomes in the haploids. As a result, the DH line contains two sets of chromosomes that are exact replicas of each other [2,3]. The DH technology facilitates the rapid attainment of genetically homozygous, ~99% pure inbred lines with distinctive characteristics, uniformity, and stability from heterozygous germplasm in a single generation [4], which helps to achieve rapid varietal registration and secure breeders’ property [5]. The DH technology has been widely adopted as an effective alternative to conventional recurrent self-breeding. It is being used for upkeeping germplasm, discovering the genetic diversity in maize, and expanding the genetic base of exclusive top germplasm through novel variations [6,7,8]. During the development of conventional parental lines, plant breeders have to face different and complex challenges of segregation, linked undesirable traits, uncontrolled morphological variation, unpredicted traits, and an ambiguous selection of recessive unexpressed desired traits. During DH line development, higher heritability and genetic variance facilitate proper selection [5]. The DH technology overcomes obstacles during recurrent self-breeding and provides an efficient, cheaper, timely, and less laborious substitute. Doubled haploids achieved after doubling chromosomes are considered novel, highly uniform genotypes because of homozygosity at all loci [9]. During the last decade, the DH technology opened new avenues of research; for example, genetic engineering [10], marker-trait associated studies [11], genomics and transformation target [12], genome mapping [13], valuation of quantitative trait loci (QTL) × environmental interactions with precision [4], and QTL mapping [14]. The foremost applied approaches used in breeding to develop DH lines include androgenesis, wide hybridization, and gynogenesis [15].

Haploid plants are generally sterile [16], and their fertility can be restored spontaneously or/and by chromosomal doubling agents. Among popular chromosome doubling agents, colchicine is preferred over nitrous oxide (N_2_O) and antimitotic herbicides, such as benzamides (pronamide), dinitro–anilines (oryzalin and trifluralin), carbamates (chlorpropham and isopropyl N-3-chlorophenyl carbamate), and phosphor–thioimidates (aminoprophosmethyl/APM) due to its availability and well-developed protocols. These herbicides stop mitosis and cell division, leading to chromosome doubling [17], like colchicine [18,19]. In contrast, higher concentrations of trifluralin, APM, and oryzalin revealed equal results of chromosome doubling by colchicine in wheat during in vitro studies [9]. In maize, flufenacet, oryzalin, and trifluralin were assessed for double haploid production in vitro [20] and in vivo [21], but found unimpressive in terms of the combinations applied and concentrations used [22]. A high concentration of oryzalin significantly decreased callus regeneration; whereas, at a low concentration, oryzalin has been found ineffective regarding chromosome doubling [23]. Oryzalin had a lower survival rate than colchicine and APM but had better chromosome doubling rates than colchicine in maize [24]. Another alternative to colchicine developed by [25] employs N_2_0, which is considered a comparatively safe and effective alternative [22], but the cost of N_2_0 pressure chamber is relatively higher [25,26]. N_2_0 exposure at 0.6 MPa (MegaPascal) on maize at V3–V8 growth stage revealed even potential regarding the doubling of chromosomes, but treatment of adult plants requires big chambers and pots for growth [26].

Colchicine is far more toxic than pronamide, trifluralin, N_2_O, APM, and oryzalin [22,24,25], requiring careful handling and disposal [16,21]. “The reported oral LD50 (lethal dose, 50%) values for pronamide range from 5620 to 8350 mg/kg, oryzalin 5000 (mg/kg), trifluralin >10,000 mg/kg, colchicine 5.8 mg/kg and APM 309 mg/kg in rats. However, the LD50 for PTX solid dispersion stated 160 mg/kg, which is ~28 times higher than colchicine.” Colchicine expresses low affinity with plant microtubules, which means required in higher concentrations [27,28]. However, it has revealed a high affinity with microtubules of vertebrates [17]. Recently, safer alternatives like N_2_0 and anti-mitotic herbicides with equal efficacy of doubling chromosomes to toxic colchicine have been acknowledged. However, some basic limitations confine their usages, such as cost, availability, higher doses or concentrations, and unregistered sales, such as pronamide and APM in most African and Asian countries, limit their easy accessibility. The availability of such chemicals in high-quality pure grade is more expensive than colchicine and is limited [5]. No doubt these antimitotic herbicides (Pronamide, APM, trifluralin, and oryzalin) are comparatively less toxic but do not meet the expectations to reduce the cost per DH line. Paclitaxel (PTX) is an effective antineoplastic agent, originally extracted from the bark of *Taxus brevifolia*. Although it was generally considered a particular metabolite of *Taxus* spp., it was recently found in hazel cell cultures [27]. It is an intravenous injection to treat several cancers, including breast, cervical, and ovarian. The World Health Organization (WHO) has enlisted it among essential medicines due to its antitumor activity. According to Tel-Aviv University Safety Unit issue “Standard Operating Procedure for Paclitaxel (Taxol) in Animals,” it is considered a cytotoxic drug with no set safety standards of exposure. According to medical opinion, reducing exposure as much as possible is the best safety approach. Due to its lower solubility in water, a formulation is prepared by dehydrated ethanol (50:50, *v*/*v*) and cremophor EL, known as “Taxol” [27]. PTX is an antimitotic drug, which interferes in the normal function of microtubules, defects in chromosome segregation, spindle assembly, and cell division. Chromosomes do not achieve spindle configuration in metaphase, block further progress in mitosis, prolong cell-programmed death, or reverse the quiescent stage (G0) phase without cell division of the cell cycle. PTX with a functionary mechanism primarily inhibits the dynamics of microtubule spindle, deoxyribonucleic acid (DNA) repair, and control cell proliferation [29]. Microtubules consisting of two similar polypeptides (α and β tubulin dimers) are the key component of the cytoskeleton and mitotic apparatus in eukaryotic cells [30]. PTX can selectively bind to the subunit β of tubulin proteins and promote their polymerization and assembly. This polymerization consumes intracellular tubulin, stops the function of the cell, averts the formation of the spindle, causes profound mitotic arrest at G2/M phases, and eventually terminates mitosis [31]. It is also known as a cytoskeletal drug, which targets tubulin. It stabilizes the microtubule polymer and shields it from disassembly, whereas colchicine obstructs the microtubule assembly. It has been found to suppress detachment of microtubules from centrosomes, a normal process triggered during mitosis. In contrast to traditional anticancer drugs, PTX neither affects the synthesis of DNA and ribonucleic acid (RNA) in cancer cells nor damages DNA molecules [32].

Taurine is a non-essential amino acid containing sulfur called 2-amino-ethane-sulfonic acid [33]. It is a beneficial anticancer agent that inhibits reactive oxygen species (ROS) buildup in the tumor cells. It stops the advancement of cancer [34], maintains the concentration of calcium and stability of membranes, and sustains the process of phosphorylation [35,36]. It stimulates tumor suppressors p53 and phosphatase and tensin homolog (PTEN) [37]. Fluorescence studies revealed that taurine has a significant binding affinity with cyclin-dependent kinase 6 (CDK6), which is connected with cyclin partner and initiates a critical role in early phases of cancer development [38]. The muscle toxicities produced during chemotherapy in cancer patients can be reduced by taurine [39]. Taurine has an inhibitory concentration 50 (IC50) value equal to 4.44 μM and, according to an enzyme-inhibition assay, is considered a good inhibitor to treat cancers directed by CDK6 [38]. The supplemental taurine dose helps to protect C2C12 myoblasts against decreasing cell viability and moderately conserve the myotube differentiation capability of cisplatin-impaired myoblasts [39].

Caffeine can increase the early doubling of chromosomes and haploid plants, the androgenic induction, and spontaneously produce fertile plants [40]. It can impact cell division and phragmoplast microtubules during cytokinesis, observed by [41] in suspension-cultured cells, BY-2, of tobacco (*Nicotiana tabacum* “Bright Yellow 2”). Caffeine did not significantly affect the doubling of chromosomes in wheat anther culture. However, several caffeine treatments reported a higher frequency of chromosomes doubled plants [40]. Therefore, we decided to assess the efficacy of caffeine in combination with taurine for a higher chromosome doubling rate in our study.

Continuous efforts proceed to optimize cheaper, innovative, non-toxic, or less toxic chemical agents to improve chromosome-doubling efficiency with higher survival rates than colchicine. To meet this challenge, we designed an advanced study to explore the ability of PTX and caffeine–taurine (CAF–T) to induce chromosome doubling in maize haploids. A systematic approach was adopted, and multiple treatments with different processing times were designed based on replicated field trials to study the chromosomal doubling efficacy of PTX and CAF–T versus colchicine. Moreover, the impact of PTX versus colchicine on the morphology and physiology of haploid plants was assessed and analyzed through seed germination, photosynthetic pigments, plant and ear height, number of silks/ears, and plant weight.

## 2. Results

### 2.1. Experiment 1

#### Field Assessment-Based Comparative Efficiencies in Different Treatments of PTX Application

Under different concentrations and processing periods, the survival rate (SR), doubling rate (DR), and actual doubling rate (ADR) of PTX-treated haploid seeds and seedlings were analyzed. In addition, anthers emergence rate (AER), partial fertility rate (PFR), complete fertility rate (CFR), and total fertility rate (TFR) were also calculated based on four different types of tassels (Figure 1A) usually observed in the DH nursery field. The SRs were decreased with the increasing concentration and the processing time in both treatments of the seed-soaking method (M1) and seedling-immersion method (M2). While comparing the SR of seeds versus seedlings, M1 treatments (Figure 2A) were higher than M2 treatments (Figure 2B), which concluded that both coleoptile and root tips were more sensitive and affected by the chemical treatments. Hence, M1 is a better option than M2 to achieve higher SR. 

In all treatments of M1, the highest DRs induced by PTX were 19.8% and 21.2% in the condition of 100 μM for 16 h and 24 h, respectively. However, both conditions for DR, ADR, PFR, CFR, and TFR were determined to be non-significantly different to each other. The M2 in PTX produced the highest DR of 54.3% and TFR of 75.2% in the condition of 400 μM for 16 h, followed by another treatment that produced a DR of 46.9% and TFR of 59.4% in the condition of 100 μM for 8 h. However, both conditions were statistically non-significant regarding ADR (42.1% and 42.2%, respectively). While comparing both M1 versus M2, M2 is a better line not only to yield higher DR (Figure 2C,D) but also ADR, AER, PFR, CFR, and TFR than M1 (Figure 2E–N). AER, PFR, and CFR induced by PTX were photographed (Figure 1B). The pollen fertility induced by PTX in haploid plants is shown in Figure 1D.

### 2.2. Experiment 2

#### 2.2.1. Field Assessment-Based Comparative Efficiencies in Different Treatments of CAF-T Application

In Experiment 2, using different treatments and processing times, CAF–T application to maize haploid seeds was assessed through SR, DR, ADR, AER, PFR, CFR, and TFR (Figure 3A–G). The highest DR and TFR produced were 20.9% and 28.1%, respectively, in caffeine 2 g/L + taurine 12 g/L for 16 h (Figure 3B,G). However, the SR revealed by this treatment was 95.8% (Figure 3A), disclosing non-significant difference with control (CK). The SRs decreased non-significantly with the increasing concentration and the processing time in T1–T6 (except T5) of CAF–T versus CK. Hence, CAF–T disclosed its safe use for maize seeds. CAF–T did not show doubling efficiency in treatments of 4 g and 6 g, 16 h and all 24 h treatments (Figure 3B). However, AER was higher than doubling treatments (T1–T6) (Figure 3D), which might reveal the impact of higher concentrations of CAF–T and longer time exposure. AER, PFR, and CFR induced by CAF–T are displayed in Figure 1C. The pollen fertility induced by CAF–T in haploid plants is shown in Figure 1F.

#### 2.2.2. Field Assessment-Based Comparative Efficiencies in Different Treatments of Colchicine Application

Two treatment methods, i.e., M1 and M2, were employed using different colchicine concentrations and processing time to calculate SR, DR, ADR, AER, PFR, CFR, and TFR (Figure 4A–N). In M1 and M2, the survival rates decreased with the increasing concentration and processing time. However, M1 treatments revealed higher SR than M2 treatments (Figure 4A,B). In M1, the highest DR induced by colchicine was 29.6%, with SR 87.7% in the condition of 2.0 mM for 8 h treatment, disclosing ADR 26.0% (Figure 4E).

In M2, the highest DR induced by colchicine was calculated 43.8% with SR 72.9% in the condition of 0.5 mM for 24 h treatment (Figure 4B,D), revealing ADR 31.9% (Figure 4F). Hence, M2 was found to be better than M1. Another treatment of M2 in the condition of 2.0 mM for 8 h revealed the highest AER and TFR 35.9% and 62.1%, respectively (Figure 4H,N) among all treatments. However, DR was only 26.2%, indicating the impact of sectoral diploidization.

### 2.3. Comparative DH Seed Quantity Produced by PTX, CAF-T versus Colchicine

In addition to fertility rates, the number of seeds on DH ears subsequent from each treatment was widely variable (Appendix A). Mostly plants produced ≤ 5 seeds/ear. However, maximum number of seeds produced by PTX (M2), colchicine (M2), and CAF–T (M1) were counted as 115 (800 µM, 16 h), 95 (0.5 mM, 16 h), and 38 (1 g/L, 16 h), respectively (Figure 1E(a–c)). The following treatments of M1 treated by PTX produced 7 seeds (100 µM, 16 h), 13 seeds (100 µM, 24 h), 52 seeds (200 µM, 16 h), and 53 seeds (400 µM, 24 h; Figure 1E(d)) as well. Hence, PTX and CAF–T produced a comparatively higher seed count than standard colchicine treatment M2 (1.5 mM, 8 h) [16,42], which produced 21 seeds (Figure 1E(e)).

### 2.4. Experiment 4

#### Large-Scale (LS) Field Efficacy-Based Comparative Studies for Validation of the Best Treatment of PTX for DH Production Pipeline

To establish the efficacy of PTX versus colchicine, 1478 putative haploid seedlings from two populations were used in this experiment. PTX treatment presented consistent and significantly better SR, ADR, and TFR in both genotypes as compared to colchicine treatment (Figure 5A,B). However, DR and CFR were significantly higher in PTX than colchicine, and AER was significantly lower in PTX than colchicine in variety 1 (V1). The CFR in colchicine treatment was significantly higher than PTX, but AER presented by PTX treatment was significantly higher than standard colchicine treatment in variety 2 (V2). However, DR was non-significantly different in both PTX and colchicine in V2.

### 2.5. Experiment 5

#### 2.5.1. Morphological Studies

##### Comparative Plant and Ear Height of Treated Maize Plants by PTX versus Colchicine and CK

The plant and ear height were significantly lower for both PTX and colchicine than CK in both the genotypes. However, PTX treatment was significantly better than colchicine treatment for both plant and ear height (Figure 6A and Figure 7A,B). Colchicine-treated seedlings exhibited delayed growth compared to PTX.

##### Comparative Silks Number/Ear Treated by PTX versus Colchicine and CK

The number of silks/ears was significantly lower for both PTX and colchicine than CK treatment in both genotypes. However, PTX treatment was significantly better than colchicine treatment for the number of silk/ear (Figure 6B and Figure 7C), which helps infer that PTX lower toxicity than colchicine will ensure a greater number of D0 seeds as proved above as well (Figure 1E). The count of silks per ear for 20 plants ranged between 220 and 255, 190 and 225, and 252 and 280 for PTX, colchicine, and CK, respectively, for V1. The count of silks per ear for 20 plants ranged between 227 and 270, 175 and 236, and 263 and 320 for PTX, colchicine, and CK, respectively, for V2 (Appendix A).

##### Comparative Plant Weight and Root Growth Treated by PTX versus Colchicine and CK

Our experiment measured plant weight significantly less for PTX and colchicine than CK in both genotypes. However, PTX treatment was significantly better than colchicine treatment for plant weight (Figure 7D), which helps to deduce that PTX produced fewer toxic effects on plant growth. Plant weight measurements per plant ranged between 95.52 and 110.11 g, 63.92 and 85.56 g, and 129.95 and 147.54 g for PTX, colchicine, and CK, respectively, for Genotype V1. Plant weight measurements per plant ranged between 116.40 and 130.74 gm, 97.520 and 107.62 gm, and 140.12 and 160.24 gm for PTX, colchicine, and CK, respectively, for Genotype V2 (Appendix A). Moreover, root growth was better for PTX than colchicine (Figure 6C).

#### 2.5.2. Physiological Studies

##### Comparative Photosynthetic Pigments of Treated Seedlings by PTX versus Colchicine and CK

The chlorophyll *a* (*Chl_a_*), chlorophyll *b* (*Chl_b_*), total chlorophyll (*Chl_T_*), and carotenoid (*C_x_*) contents were significantly lower for both PTX and colchicine than CK in both the genotypes. However, PTX treatment was significantly better than colchicine treatment for *Chl_a_*, *Chl_b_*, *Chl_T_* (Figure 7E,F), and *C_x_* (Figure 7G).

##### Comparative Germination (%) Impact on Treated Seeds by PTX, CAF–T versus Colchicine

Moreover, to SR, we studied the effects of PTX and CAF–T versus colchicine on seed germination. However, the impact of colchicine on treated haploid maize seeds has not been reported. We selected the induced genotype (GO927 × 986) with Stock 6 inducer for this experiment because of its higher germination of about 97%.

Germination (%) Impact on Treated Seeds by PTX

Across all treatments (T1 to T12) of PTX, germination percentage decreased with increasing concentrations and processing time (Figure 8A). PTX-treated seed germination percentage was found to range from 95.7% to 86.8% across all respective seed-soaked treatments. PTX showed a non-significant effect on seed germination percentage from T1–T6 versus CK.

2.Germination (%) Impact on Treated Seeds by CAF–T

CAF–T-treated seed germination percentage was found between 96.2% and 77.8% across all respective seed-soaked treatments. CAF–T exhibited a non-significant impact on seed germination percentages on T1 and T2 versus CK (Figure 8B).

3.Germination (%) Impact on Treated Seeds by Colchicine

The colchicine-treated seed germination percentage ranged from 90.8% to 80.5% across all respective seed-soaked treatments (Figure 8C). In contrast, no colchicine treatment showed a non-significant effect versus CK; therefore, colchicine proved its more toxic effects on seed germination.

In conclusion, PTX revealed a lesser impact on seeds germination percentage versus colchicine.

### 2.6. Experiment 6

This experiment aimed to validate the chromosomes doubling induced by PTX and CAF–T in the field studies through the microscope.

#### 2.6.1. PTX Induced Chromosome Doubling Signals Detected by Fluorescence in Situ Hybridization (FISH) Using Knob-2 as Probe

##### PTX Induced Chromosome Doubling by Seed-Soaking Method (M1)

The M1 revealed DR ranging from 60% to 80% across all treatments of PTX based on chromosome doubling count under the microscope using root tip cells.

Maximum DR in M1 achieved 80% by these treatments, i.e., PTX 800 μM 16 h, 24 h, and 48 h: PTX 200 μM 48 h (Figure 9A).

##### PTX-Induced Chromosome Doubling by Root-Immersion Method (M3)

The M3 showed DR ranging from 56.7% to 70.0% across all treatments of PTX. A maximum DR (70%) was achieved through M3 at concentrations of PTX 400 μM and 800 μM for 16 h (Figure 9B).

In conclusion, M1 exhibited better frequency and DR than M3. In addition, we noticed 48 h treatments were not exceptionally better than 24 h treatments both in M1 and M3. Therefore, it helps deduce that 48 h treatments cannot improve DR further but decrease SR. The CK treatment showed a spontaneous doubling ratio of about 3%. Hence, PTX proved effective in doubling chromosomes, as shown in Figure 10A,B.

### 2.7. Experiment 7

#### 2.7.1. CAF–T-Induced Chromosome-Doubling Signals Detected by FISH Using Knob-2 as Probe

##### CAF–T-Induced Chromosome Doubling by Seed-Soaking Method (M1)

The M1 revealed DR ranging from 53.3% to 86.7% across all treatments of CAF–T based on chromosome doubling count under the microscope using root tip cells. The maximum DR in M1 was achieved at 86.7% by caffeine 2 g/L + taurine 12 g/L, 16 h (Figure 9C).

##### CAF–T-Induced Chromosome Doubling by Root-Immersion Method (M3)

The M3 disclosed DR ranging from 33.3% to 53.3% across all treatments of CAF–T. The maximum DR (53.3%) was achieved by M3 at a concentration of caffeine at 2 g/L and taurine at 12 g/L for 8 h; caffeine at 6 g/L and taurine at 36 g/L for 8 h; caffeine at 4 g/L and taurine at 24 g/L for 24 h; caffeine at 6 g/L and taurine at 36 g/L for 24 h; no further increases in ratios were observed (Figure 9D).

In conclusion, CAF–T-based M1 exhibited better frequency and DR than M3. In addition, we observed that 48 h treatments cannot improve DR further but decrease SR. Hence, CAF–T verified its efficacy in doubling chromosomes, as shown in Figure 10C.

## 3. Discussion

No standard set criteria are used to assess chromosomal doubling agents’ efficacy. Some studies were based on the microscopic chromosomal count [43], several results were established on the basis of seed production [5,44], many experiments were based on flow cytometry [24], and numerous researchers considered pollen production [45] as an indicator of successful chromosome doubling. We generated results based on field evaluation and assessment, as well as further confirmation through the microscope. Most researchers did not bother about the mortality of treated plants with toxic chemicals. However, [46] suggested three parameters named survival rate (SR), reproduction rate (RR), and overall success rate (OSR). Basically, to determine the efficacy of chromosome doubling agent and to achieve economically cheaper DH lines, two or three basic parameters are required, i.e., SR, fertility rate, and/or seed setting. Naturally, the haploid plants are weak and prone to several abiotic and biotic stresses [47]. The poor seed setting due to any biotic and abiotic factors, such as disease, temperature/heat, sunlight, irrigation, or rainfall, etc., can impact the efficacy of the chemical agent. Hence, we concentrated on two basic parameters, i.e., the SR and fertility rate, which determine the ADR = SR × DR. The OSR suggested by [46], and the ADR indicated by [24], are similar but differ in that OSR is based on survival and seed production, whereas ADR is based on survival and fertility. To assess the fertility and efficacy of chemical agents, we further divided the fertility rate into AER, PFR, and CFR. According to [48,49], AER is an effective parameter to determine fertility in haploids, but pollen-less anthers cannot pollinate. However, it can be considered as these plants have a potential probability of being fertile, but this might be due to genotypic or treatment effects that could not double all chromosomes. TFR predicts the success probability or potential of any treatment that might be underrated due to genotypic impact, treatment, and spontaneous chromosomal doubling. The genotypic influence [48] and spontaneous chromosomal doubling [50,51,52] impact on the efficacy of artificially induced chromosome doubling have been reported.

In our experiments, the CAF–T-based M1 disclosed SR ranges from 80.4% to 99% across all treatments. The PTX-based M1 and M2 showed SR ranges from 73.4% to 98.5% and 56.4% to 90.1%, respectively, whereas colchicine-based M1 and M2 revealed SR varies from 69% to 90.8% and 29% to 82.6%, respectively, across all treatments. After treatment with colchicine, the established seedlings can be only 40 to 80%, in which 10–30% diploids or false positives may be observed in the field [53]. The higher SR of PTX-treated maize seedlings revealed weaker inhibitory action of PTX than colchicine. Colchicine has a toxic effect that depends upon the germplasm’s background and the treatment method/process, leading to the loss of many treated seedlings during chromosomal doubling. In conclusion, M1 proved better SR than M2 in our studies and endorsed [24]. The CAF–T disclosed the highest SR and declared it the safest chromosome-doubling chemical for maize haploid seeds compared to PTX and colchicine.

The spontaneous doubling rate was 1.4% in Experiments 1, 2, and 3, which was very low and almost equal to the reported 1.5% [24]. However, germplasm used in Experiments 4, 6, and 7 revealed a spontaneous doubling rate of 3.1% (V1, Exp. 4), 5% (V2, Exp. 4), and 3.3% (Exp. 6 and 7), which was significantly much lower in CK than treated treatments by PTX, colchicine, and CAF–T. Spontaneous fertile plants from diverse maize germplasm groups revealed DR ranging from 0 to 16.7% in Iodent heterotic groups, Lancaster, and intra-pool crosses from Stiff Stalk [48]. The spontaneous fertility depends upon the maize genotypes [48,50,53].

Overall, M2 was more effective than M1 in achieving higher DR in our field experiments because many stem cells were exposed to chemical treatments leading to tassel fertility. M2 also proved better in tetraploids than M1 [54]. However, M1 was observed to be more effective in Experiments 6 and 7 than M3 because a considerable number of seed cells are in the phase of mitosis [24]. The difference in results of field experiments 1 and 2 versus microscopic experiments 6 and 7 is based on two reasons: seeds of two different genotypes were used in these experiments, and, in the microscopic Experiments 6 and 7, root cells were studied under the microscope. The comparative study of M2 versus M3 using antimitotic herbicides [21,46] and colchicine revealed poor success for M3 in field experiments [19,55,56]. Both immersions of roots and crown region exposing the shoot apical meristem cells were recorded as being more successful in achieving a higher OSR [5]. In our Experiments 1 and 3, overall, M1 revealed poor results compared to M2, similar to the studies conducted by [25,46]. Chalyk [45] also reported no fertility using M1. Successful colchicine treatment depends on exposure time, concentration, and contact with meristematic cells [54].

In M1, PTX, colchicine, and CAF–T exhibited an AER range from 4.9% to 16.6%, 4.7% to 32.8%, and 4.5% to 12.4% across all treatments. In M2-based experiments, PTX and colchicine displayed AER range from 5.9% to 28.7% and 7.8% to 35.9%, respectively, across all treatments. In diverse China and US germplasm, the reported AER with significant genetic variance varied from 9.8 to 89.8%. [51].

In the LS field efficacy-validation experiment, PTX and colchicine-treated maize seedlings showed an ADR of 37.7% and 19.2%, respectively, in V1, 20% and 14.5%, respectively, in V2, whereas [24] reported 20.64% of ADR by colchicine. The DR and AER are inversely proportional to each other in both genotypes because if DR is higher, AER may be reduced. However, if AER is higher, DR may be lower due to sectoral diploidization. According to [53], colchicine treatment may or may not ensure that all cells’ chromosomes are doubled, which is known as sectoral diploidization. The sectoral diploidization effect varies from genotype to genotype and colchicine application.

Haploid plant tassels cover a wide range of fertility, from one or two anthers shedding pollen to the complete tassel [22]. Therefore, we divided the fertility or DR into two types, i.e., PFR and CFR, to assess the doubling efficacy of the chemical agents. In LS, PTX showed PFR and CFR 29.4% and 24.5%, respectively; in V1, 16.6% and 11.2%, respectively, in V2, whereas colchicine showed PFR and CFR 17.1% and 20.7%, respectively, in V1, and 13.9% and 16.5%, respectively, in V2. Hence, PTX showed more CFR in V1 as compared to colchicine. There is a probability that 0–40% of colchicine-treated plants can have both silks and the ability to produce pollen (DR) for successful pollination among the remaining true haploids [53]. In Experiments 1 and 3, PTX and colchicine based M2 showed a maximum of DR 54.3% and 43.8%, respectively. In LS, PTX and colchicine showed DR 53.9% and 37.8% in V1, respectively, whereas PTX and colchicine-treated seedlings showed DR 27.8% and 30.4% in V2, respectively. Next, [24] calculated the highest doubling rate of 29.7% using colchicine, whereas it was reported in the same publication the colchicine chromosome doubling rates by studies of Wen’s and Liu’s 48.35% and 23.0%, respectively.

Colchicine, PTX, and CAF–T-treated M1 revealed a maximum DR of 21.2%, 29.6%, (0.08% for colchicine concentration for 8 h), and 20.9%, respectively, but colchicine induced 18% of DR at 0.06% colchicine for 12 h treatment [57,58,59,60]. CAF–T did not show doubling efficiency in treatments of 4 g and 6 g, as well as for 16 h and all 24 h treatments, which might reveal the impact of higher concentrations of CAF–T and longer time exposure. In addition, Ref. [61] also suggested caffeine as a fertility-inducing agent by root dipping of wheat with best-achieved results at the application of 3 g/L for 24 h.

PTX and CAF–T produced comparatively higher seed counts than colchicine. According to [53], only 30–50% of colchicine-treated haploid plants produce seeds. Another fact is that plants treated with chromosome-doubling chemicals like colchicine produced fewer seeds than spontaneously doubled haploid plants [62]. Toxic chemicals like colchicine further stress weak haploid plants [22]. Hence, we can conclude that bio-safe or less toxic chemicals are suitable for a higher seed setting as in our experiment PTX, and CAF–T produced more seeds per D1 ear. D1 ears’ ability to produce seeds varies widely. However, on average, four seeds per D1 ear by colchicine have been reported [48]. At CIMMYT, according to unpublished data, 40–60% of the D0 plants treated by colchicine produced more than 25 seed grains depending on the season, whereas the rest exhibited poor seed setting [22].

The colchicine effects on the morphology and physiology of treated maize haploid seeds or seedlings have not been reported. However, only SR of both maize haploid seeds and seedlings have been widely reported. In addition to SR, PTX unveiled less impact on seeds germination percentages as compared to colchicine. Severe effects of different colchicine concentrations were also reported in *Phlox drummondi* [63]. In *Prunella vulgaris* and *Dendrocalumus brandisii*, a higher colchicine concentration with longer processing times decreased the germination of seeds [64,65]. Colchicine-treated okra seeds’ germination was calculated 79.3%, which is significantly lower than untreated plants at 94.8% [66]. We found PTX treatment to be better than colchicine for plant weight, the number of silks/ear, and plant and ear height in both the genotypes. Then, Ref. [67] studied the impact of colchicine on plant height in Jimsonweed (*Datura stramonium* L.), which decreased linearly by increasing incubation time and the concentration of colchicine. A decrease in plant height in *Abelmoschus esculentusalso* (okra) and *Phlox drummondi* was measured due to colchicine effects [63,66]. Colchicine disclosed more adverse effects on the growth of seedlings than on root growth and seed germination in *Dendrocalumus brandisii*. Comparatively, colchicine revealed more negative effects regarding the degradation of sucrose than starch during the germination of seeds and seedling growth [65].

Chlorophyll is the most essential pigment for photosynthesis because it largely determines the process’s capacity and plant growth. The *Chl_T_* directly impacts the plants’ photosynthesis capacity [68]. The carotenoids have three considerable roles in photosynthesis: (a) A pigment to harvest accessory light, (b) Wavelengths range extension to facilitate light for the photosynthetic process, (c) Protection of achlorophyllous pigments from photo-destructive reaction [69]. Damage in chlorophyll and carotenoid contents decreases the efficacy of plants. Therefore, PTX damage to chlorophyll and carotenoids were significantly lesser than toxic colchicine in our study. Single-Stranded Oligonucleotides (SSONs) serve as probes that are a popular method for chromosome detection, painting, and identification [70] because of their versatility, high resolution, sensitivity, and cost effectivity [71]. SSON probes are used after labelling with detectable stable signals, and in our experiments, these signals confirmed the efficacy of PTX and CAF–T regarding the doubling of chromosomes.

## 4. Materials and Methods

### 4.1. Germplasm, Chemicals, Instruments and Experimental Locations

Maize haploids seeds were produced at the College of Agriculture, Guizhou University (26°25′21″ N, 106°40′09″ E), China. Haploid seeds were separated visually from diploids by an *R1*–*nj* anthocyanin marker expression on kernels. The required sample sizes were obtained from 16,400 putative haploid seeds for all experiments in this embodiment. The details of genotypes and experimental locations are demonstrated in Appendix A.

All treatments were replicated thrice in Experiments 1, 2, and 3. Each replication has an average of 70 putative haploids—some treatments have more than 70 seeds/seedlings to ensure equal stand with lower-dosed treatments because higher doses affect plant stands. All three replications of each treatment were blocked in a randomized complete block design (RCBD), including Experiment 4. PTX was purchased from Peptide Biotechnology Co. Ltd., Xi’an, China, CAF–T from BANNY DEER at Wuhan East–Lake High Tech Region, China, and colchicine (98%) from Shanghai Macklin Biochemical Co. Ltd., Shanghai Chemical Industry Park, Shanghai, China. MAPADA P1 UV–Visible Spectrophotometer was purchased from Shanghai Mapada Instruments Co., Ltd. Shanghai, China.

### 4.2. Treatments & Concentrations

#### 4.2.1. Experiment 1

The putative maize haploid seeds and seedlings (including roots and stem) were treated with PTX at 28 °C for 8, 16, and 24 h. The stock solution concentrations of PTX were used for chromosomal doubling, including 100 μM (0.085 g/L), 200 μM (0.170 g/L), 400 μM (0.341 g/L), and 800 μM (0.682 g/L). PTX is highly lipophilic and has a poor water solubility of less than 0.1 μg/mL, seriously affecting its bioavailability [72]. The method adopted to make PTX soluble was suggested by [73], with a surfactant tween-80 at the ratio of 1:3 diluted in required volume in addition to 2% DMSO (same for each treatment) as a penetrating agent; this was adjusted by distilled deionized water (ddH_2_O).

#### 4.2.2. Experiment 2

The putative maize haploid seeds were soaked in CAF–T at 28 °C for 8, 16, and 24 h. The concentrations of CAF–T were used for chromosomal doubling, including caffeine at 1 g/L and taurine at 6 g/L, caffeine at 2 g/L and taurine at 12 g/L, caffeine at 4 g/L and taurine at 24 g/L, and caffeine at 6 g/L and taurine at 36 g/L, with 0.1% DMSO, which was adjusted by ddH_2_O.

#### 4.2.3. Experiment 3

The putative maize haploid seeds and seedlings (including roots and stem) were treated with colchicine at 28 °C for 8, 16, and 24 h. The different concentrations of colchicine were used for chromosome doubling, including 0.5 mM (0.2 g/L), 1 mM (0.4 g/L), 1.5 mM (0.6 g/L), and 2 mM (0.8 g/L) [24] with 0.1% of DMSO, which was adjusted by ddH_2_O.

#### 4.2.4. Experiments 4 and 5

LS field efficacy-based comparative study was designed to further authenticate the best treatment (concluded in Experiment 1) of the new chemical, i.e., PTX (400µM and 2% of DMSO and 1.02 mL/L of tween-80 for 16 h). The morphological and physiological measurements were also taken. The putative maize haploid seedlings from two populations, i.e., V1 and V2, were used to compare PTX with colchicine treatment (2 mM and 0.1% of DMSO for 8 h treatment) as the best treatment in Experiment 3 regarding TFR and also suggested by [5] and control (ddH_2_O).

#### 4.2.5. Experiments 6 and 7

To verify the doubling of chromosomes under the microscope, the same concentrations of PTX and CAF–T (see Section 4.2.1) were used for seed soaking and root immersion for 8, 16, 24, and 48 h.

### 4.3. Methods

#### 4.3.1. Seed-Soaking (Method 1/M1)

The putative maize haploid seeds were first soaked in ddH_2_O for 6 h at 28 °C, and then soaked in three chemical agents, i.e., PTX, CAF–T, and colchicine for 8, 16, and 24 h at 28 °C. After chemical treatments, seeds were rinsed with ddH_2_O for 30 min and shifted on a wet germination paper in an incubator with a maintained temperature of 28 °C [24]. Afterwards, the seedlings of 1–1.5 cm in length were transferred to trays.

#### 4.3.2. Seedling-Immersion (Method 2/M2)

This protocol is a “standard seedling-immersion method” defined in the 1990s [19,43]. The putative maize haploid seeds were soaked in ddH_2_O at 28 °C for 6 h and shifted on a moist germination paper for 72–96 h in an incubator temperature maintained at 28 °C for germination. When the roots achieved their length up to 1 cm, a 28 °C temperature decreased to 24 °C in demand to make shoots grow. When shoots’ sizes increased up to 2 cm, the coleoptiles’ top 2–3 mm [24] and roots above 2 cm [53] were cut and dipped in distilled water for 30 min for stress recovery. Then, these seedlings were immersed in PTX and colchicine for 8, 16, and 24 h at 28 °C. Later, they were rinsed well, soaked in ddH_2_O for 30 min, and then shifted into trays for growing in the greenhouse.

#### 4.3.3. Root-Immersion (Method 3/M3)

We followed the protocol of root immersion as suggested by [24]. The roots were immersed in the agents (PTX and CAF–T) for 8, 16, 24, and 48 h. Thirty seeds and seedlings (roots only) were treated to confirm the efficacy of each treatment under the microscope using Knob-2 probe.

#### 4.3.4. Acclimatization of Seedlings in Greenhouse and Transplanting in Field

After respective treatments, seedlings were transplanted into trays filled with suitable soil composition, sand, peat, and compost with pH 6 to 6.5. Treated seedlings were grown in a greenhouse for 15 h of photoperiod during the day and maintained at about 26–28 °C and 24–28 °C at night. Nutrients and water were provided on a regular need basis. Gradually transferring the seedlings from lower light intensity to higher light intensity and higher-to-lower humidity helped acclimate the treated seedlings [9].

When survived seedlings achieved V2 leaf stage [5] more or less after 15 days (depending upon growth), they were transplanted [42,53] in the well-ploughed and prepared field. Transferring treated seedlings from the greenhouse to the field at V2 was a cautious and delicate process. Survived seedlings’ frequency mainly depends upon the treatment with chromosomal doubling agents and on growing condition, medium, genetic effect, technical equipment, and human skills [9]. The chimeric and weak maize seedlings required much-needed special care and attention, but these seedlings were not found to be useful. As a rule, these seedlings degenerated and did not survive in the greenhouse and field. Losses of seedlings lead due to hardening of seedlings [9]. The survived seedlings were grown to harvest and provided the best agronomy as recommended by [53]. Haploid plants are commonly weak [47], and chemical treatments enforce further stress. Therefore, they were handled with extreme care during and after chemical treatments and transplanting to harvesting. Pollen-producing plants were self-pollinated, and harvested seeds were visually observed at plant maturity for the marker (should have no marker color) to identify them as double haploids.

#### 4.3.5. Experiments 6 and 7

The knob-2 probe used in our experiment was developed from tandem repeats of plasmid clones. The 180-base pair (bp) knob sequence was obtained by dividing the 180 bp knob sequence into three non-overlapping parts, each containing 59 nucleotides, and then modifying the 5’ ends with fluorescein amidites (FAM). The knob-2 probe sequence and maize chromosomes were prepared according to the protocol provided by [70]. This is the knob-2 probe sequence:

GAAGGCTAACACCTACGGATTTTTGACCAAGAAATGGTCTCCACCAGAAATCCAAAAAT

The fluorescence in situ hybridization (FISH) procedure was followed as described by [74].

### 4.4. Assessment Methodology and Data Collection Procedure

#### 4.4.1. Experiments 1–4

In Experiments 1–4, the following parameters were documented: (a) Number of seeds or seedlings treated; (b) Germination percentage; (c) Number of survived Do plants at pollination; (d) Count of Do plants showed anthers only; (e) Sum of Do plants that exhibited partial fertility; (f) Number of plants presented complete fertility; (g) Number of false/wrong diploid plants; (h) Number of haploid plants. The following calculations were made in Experiments 1–4 and expressed in percentages (%).
Survival rate (SR) = count of survived plants at pollination/number of seedlings treated × 100 [46];Doubling rate (DR) = sum of plants showed partial and/or complete fertility/count of haploid survived plants at pollination × 100;Actual doubling rate (ADR) = SR × DR/100 [24];Anthers emergence rate (AER) = number of plants showed anthers only/number of haploid plants survived at pollination × 100 [48,49];Partial fertility rate (PFR) = number of plants that exhibited partial fertility/number of haploid plants that survived at pollination × 100;Complete fertility rate (CFR) = number of plants presented complete fertility/number of haploid plants that survived at pollination ×100;Total fertility rate (TFR) = AER + PFR + CFR.

In addition, seed setting data were also collected in Experiments 1–3 and expressed as follows; (a) Percentage of DH lines with 1–5 seeds; (b) Percentage of DH lines containing 6–25 seeds; Percentage of DH lines containing > 25 seeds [5]; Reproduction rate (RR); Overall success rate (OSR) [46].

Percentage of DH lines with 1–5 seeds = number of D_0_ plants/D_1_ ears produced 1–5 seeds/number of haploids survived plants at pollination × 100;Percentage of DH lines containing 6–25 seeds = number of D_0_ plants/D_1_ ears produced 6–25 seeds/number of haploid survived plants at pollination × 100;Percentage of DH lines containing > 25 seeds = number of D_0_ plants/D_1_ ears produced > 25 seeds/number of haploid survived plants at pollination × 100;RR = number of D_0_ plants produced seeds/number of D_0_ survived haploid plants at pollination × 100;OSR = number of D_0_ plants produced seeds/number of putative seeds/seedlings treated with chemical agent × 100.

#### 4.4.2. Experiment 5

##### Morphological Studies

Plant and Ear Height, Plant Weight, and Number of Silks/Ear

Five plants per repeat were randomly selected (evading three plants at each plot beginning and end to exclude any impact) from each treatment and genotype. Plant and ear height were measured randomly from haploid and diploid (false positive) plants. Plant height was measured from the plant base or soil to the last collar leaf. Ear height measurements were obtained from the plant’s base (ground/soil) to the node-bearing upper ear. Plant and ear height data were measured from three replications and analyzed as means. Cut the plants from the ground surface and weigh them on a scale in grams (g). The mean plant weight and number of silks/ears were calculated from four replications.

##### Physiological Studies

Measurement of Photosynthetic Pigments (*Chl_a_*, *Chl_b_*, *Chl_T_*, and *C_x_*)

Maize leaf samples were obtained randomly at V2 leaf stage in each sample area and sealed in aluminum foil to measure in the laboratory via MAPADA P1 UV–Visible Spectrophotometer. Five random plants were selected from four different replicated plots from each treatment and genotype and analyzed as mean *Chl_a_*, *Chl_b_*, *Chl_T_*, and *C_x_*. With the aid of liquid nitrogen, 0.2 g of fresh leaf sample were ground into a powder and mixed into 1.5 mL of 95% ethanol [75,76]. The solution was mixed well and placed in darkness for 10 min. Afterwards, the solution was centrifuged at 8000–10,000 rpm for 8–10 min and the supernatant was separated at 1.2 mL in another glass tube for the spectrophotometer. Then, 1.2 mL 95% of ethanol were employed as a blank. The collected supernatant was measured for absorbance at 665, 649, and 470 nm in the spectrophotometer. The concentrations of *Chl_a_*, *Chl_b_*, *Chl_T_*, and C_x_ were calculated in mg. g^−1^ using the following equations: *Chl_a_* = (13.95 × A_665_ − 6.88 × A_649_) × 1.5×10^−3^ ÷ 0.18,
*Chl_b_* = (24.96 × A_649_ − 7.32 ×A_665_) × 1.5×10^−3^ ÷ 0.18,
*Chl_T_* = *Chl_a_* + *Chl_b_*,
*C_x_*= (1000 × A_470_ − 2.05 × *Chl_a_* − 114.8 *Chl_b_*) ÷ 245 × 1.5×10^−3^ ÷ 0.18,

The leaf samples were obtained in such a way as to avoid unbiased results; 20 leaf samples were obtained from four replicated plots.

2.Germination Percentages

The impact of PTX and CAF–T versus colchicine on seed germination was assessed and calculated as the count of germinated seedlings/total number of treated seeds × 100.

#### 4.4.3. Experiments 6 and 7

For each maize seed or root treated with PTX or CAF–T, a slide of root cells was prepared, and 300 cells/slide were observed for maize chromosomes. Knob-2 probe signals counted under Olympus BX53 fluorescence microscope. The images were obtained with Olympus DP80 camera. The following data parameters were observed and recorded: (a) Total number of maize haploid seeds/roots were treated; (b) The number of treated seeds or roots doubled with 5–10% cells. The percentage of treated seeds/roots doubled was calculated as the sum of treated seeds or roots with cells doubled up to 5–10% and divided by the number of seeds or roots treated × 100.

### 4.5. Statistical Analysis and Graphics Improvement

In Experiments 1 to 5, the results of all three or four replications (*n* = 3 or *n* = 4) were expressed in graphs based on standard error (SE). The mean comparison was made by the “Tukey’s Honestly Significant Difference (HSD)” test at *p* ≤ 0.05, following the Analysis of Variance (ANOVA). All replicated data were analyzed via IBM SPSS Statistics 21.0 software. The graphics software known as “Origin” was applied to improve graphics (Version 2022, Origin Lab. Corporation, Northampton, MA, USA).

## 5. Conclusions

PTX produced better results than the widely used toxic traditional chemical agent (colchicine) to improve chromosome-doubling success rates and survivability. The ADR for PTX and colchicine in maize haploid seedlings were 42.1% (400 M, 16 h treatment) and 31.9% (0.5 mM, 24 h treatment), respectively. The ADR induced in maize haploid seeds by colchicine, CAF–T, and PTX were 26% (2.0 mM, 8 h treatment), 20% (caffeine 2 g/L + taurine 12 g/L, 16 h), and 19.9% (100 μM for 24 h treatment), respectively. PTX and CAF–T offer occupational health, operational ease, and disposal. PTX guarantees a cost-effective and better rate of doubling haploid plants at lower concentrations. Optimizing chromosomal doubling protocols based on less toxic or bio-safe chemicals like PTX and CAF–T will further enhance the overall efficiency of DH production. The morphological and physiological effects produced in haploid plants by PTX were significantly lower than toxic colchicine. PTX will improve the quantity of D0 lines seeds, which may exterminate the requirement to increase D1 lines seeds, thereby saving time and cost of the additional cycle to multiply seeds.

## Figures and Tables

**Figure 1 ijms-24-14659-f001:**
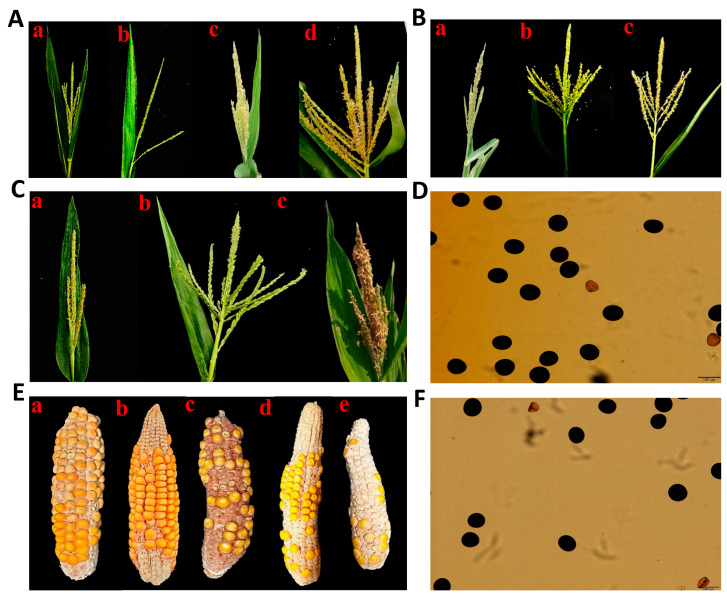
Paclitaxel (PTX) and Caffeine–Taurine (CAF-T) fertility and seed-setting efficacy. (**A**) Four types of maize tassels (**a**) Sterile, (**b**) Pollenless anthers, (**c**) Partially fertile, and (**d**) Complete fertile tassel in the DH field. (**B**) PTX-induced fertility (**a**) Anthers only, (**b**) Partial fertility, (**c**) Complete fertility. (**C**) CAF–T-induced fertility (**a**) Anthers only, (**b**) Partial fertility, (**c**) Complete fertility. (**D**) Effective pollen fertility induced by PTX. (**E**) Maximum DH seed quantity produced by (**a**) PTX seedling inversion method (M2), (**b**) Colchicine M2, (**c**) CAF-T seed soaking method (M1), (**d**) PTX M1, and (**e**) Standard colchicine treatment. (**F**) Effective pollen fertility induced by CAF-T.

**Figure 2 ijms-24-14659-f002:**
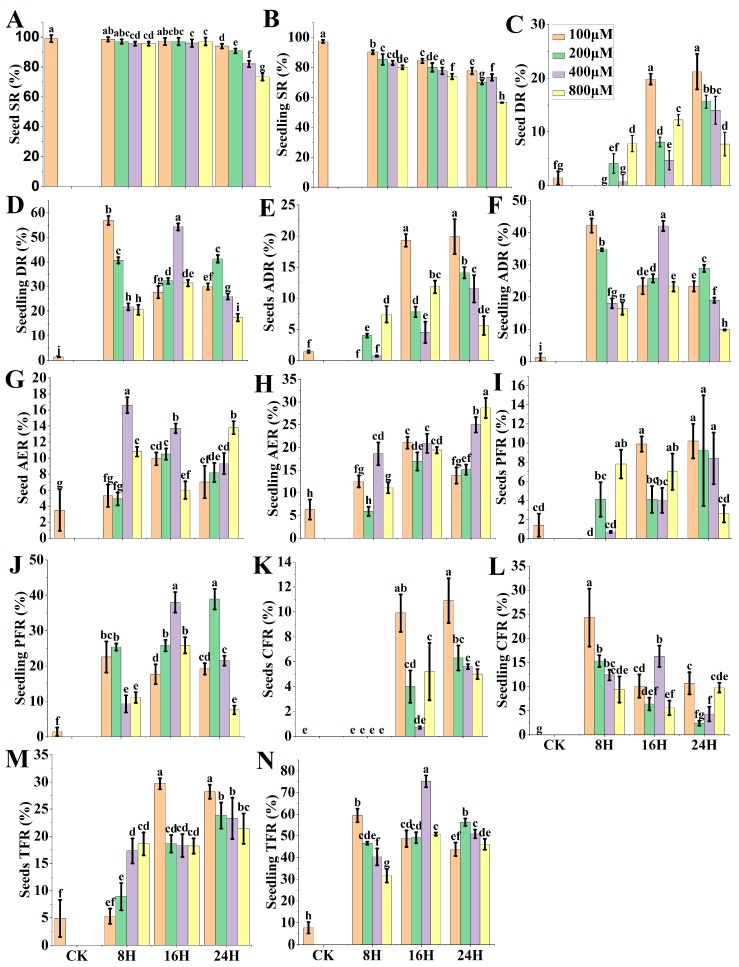
Paclitaxel (PTX) seed-soaking method (M1) versus seedling-immersion method (M2) efficacy. (**A**) Seed SR (survival rate), (**B**) Seedling SR, (**C**) Seed DR (doubling rate), (**D**) Seedling DR, (**E**) Seed ADR (actual doubling rate), (**F**) Seedling ADR, (**G**) Seed AER (anthers emergence rate), (**H**) Seedling AER, (**I**) Seed PFR (partial fertility rate), (**J**) Seedling PFR, (**K**) Seed CFR (complete fertility rate), (**L**) Seedling CFR, (**M**) Seed TFR (total fertility rate), (**N**) Seedling TFR. CK = control; this treatment is identical to all other treatments, but PTX, Di-methyl Sulfoxide (DMSO), and tween-80 solution were replaced by distilled deionized water (ddH_2_O). Different small letters on bars represent the significant differences within the treatments calculated using Tukey’s HSD test at *p* ≤ 0.05. Vertical bars on graphs indicate the standard error of the mean (*n* = 3).

**Figure 3 ijms-24-14659-f003:**
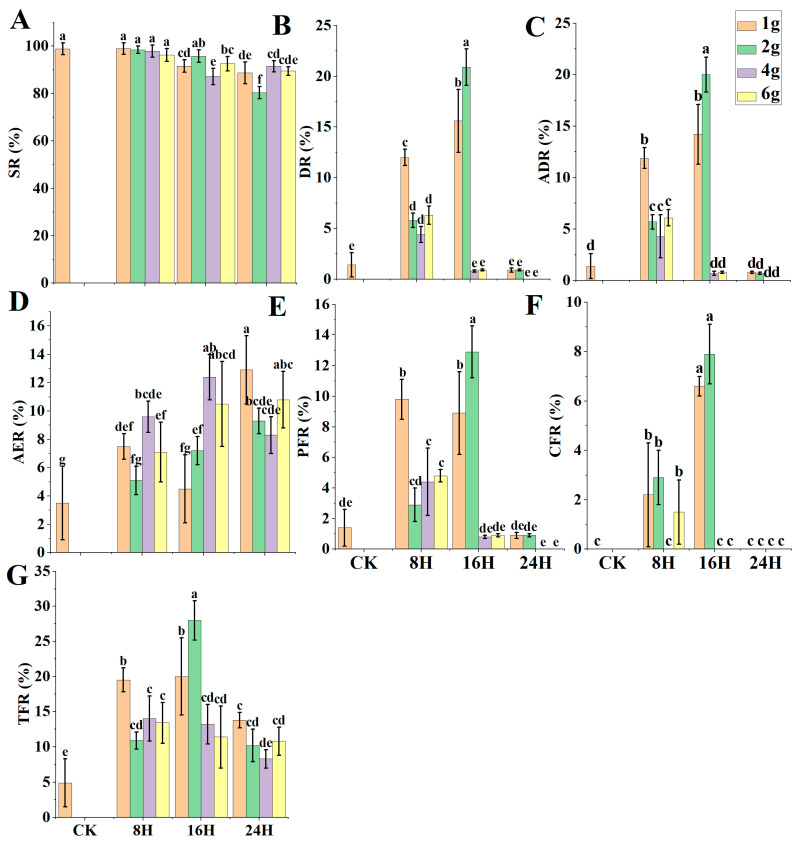
Caffeine–Taurine (CAF–T) seed-soaking method efficacy. (**A**) Seed SR (survival rate), (**B**) Seed DR (doubling rate), (**C**) Seed ADR (actual doubling rate), (**D**) Seed AER (anthers emergence rate), (**E**) Seed PFR (partial fertility rate), (**F**) Seed CFR (complete fertility rate), (**G**) Seed TFR (total fertility rate). CK = control; this treatment is identical to all other treatments, but distilled deionized water (ddH_2_O) replaced CAF–T and Di-methyl Sulfoxide (DMSO) solution. Different small letters on bars represent the significant differences within the treatments calculated using Tukey’s HSD test at *p* ≤ 0.05. Vertical bars on graphs indicate the standard deviation of the mean (*n* = 3).

**Figure 4 ijms-24-14659-f004:**
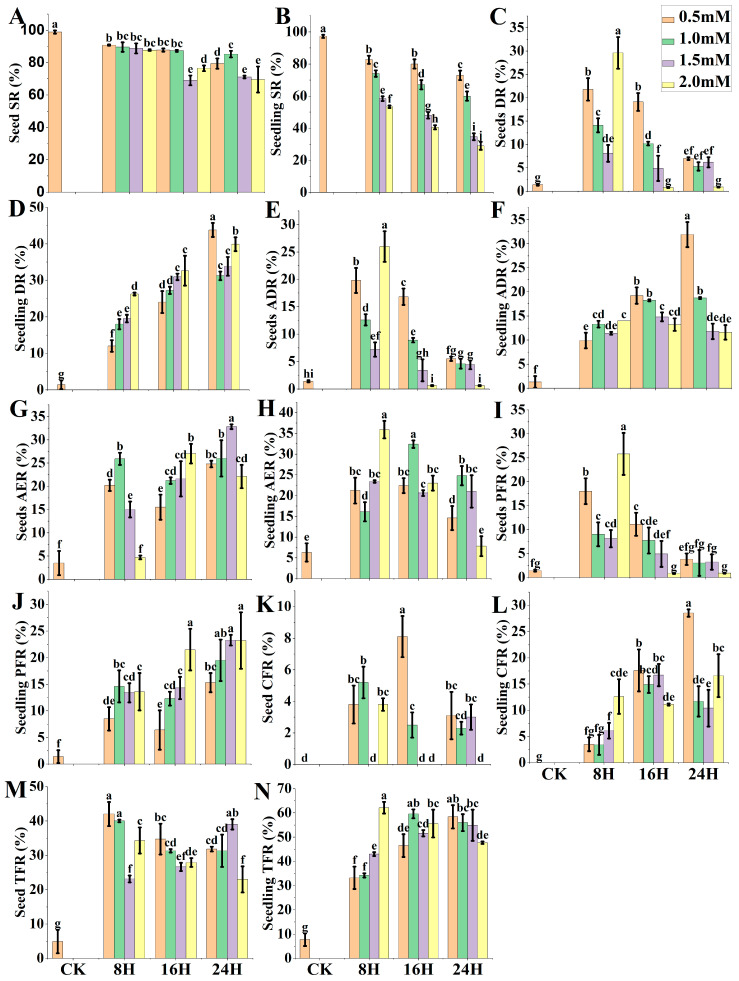
Colchicine seed-soaking method (M1) versus seedling-immersion method (M2) efficacy. (**A**) Seed SR (survival rate), (**B**) Seedling SR, (**C**) Seed DR (doubling rate), (**D**) Seedling DR, (**E**) Seed ADR (actual doubling rate), (**F**) Seedling ADR, (**G**) Seed AER (anthers emergence rate), (**H**) Seedling AER, (**I**) Seed PFR (partial fertility rate), (**J**) Seedling PFR, (**K**) Seed CFR (complete fertility rate), (**L**) Seedling CFR, (**M**) Seed TFR (total fertility rate), (**N**) Seedling TFR. CK = control; this treatment is identical to all other treatments, but distilled deionized water (ddH_2_O) replaced colchicine and Di-methyl Sulfoxide (DMSO) solution. Different small letters on bars represent the significant differences within the treatments calculated using Tukey’s HSD test at *p* ≤ 0.05. Vertical bars on graphs indicate the standard error of the mean (*n* = 3).

**Figure 5 ijms-24-14659-f005:**
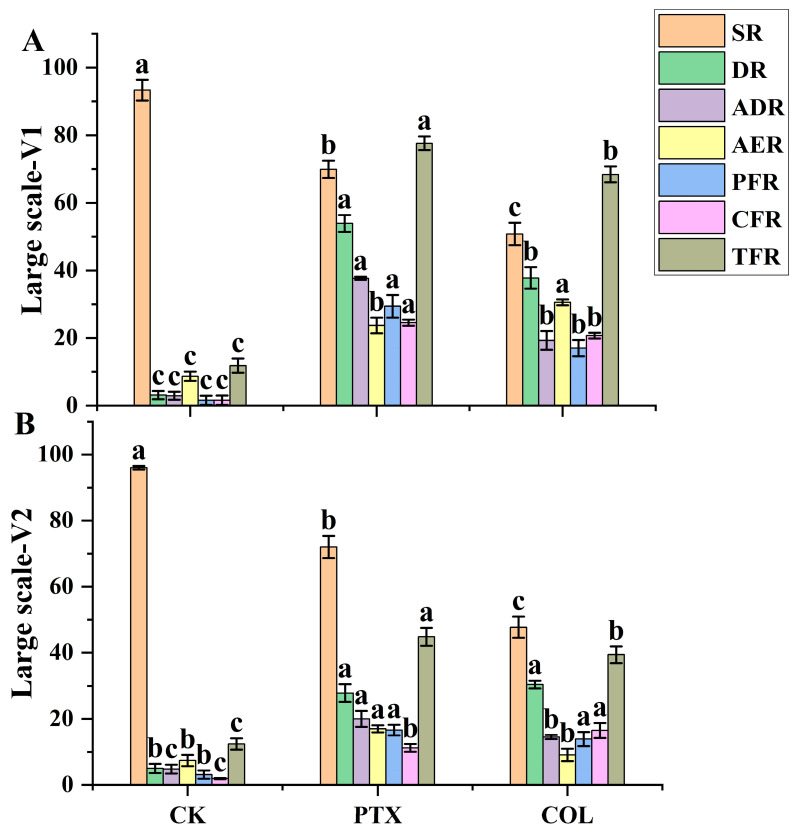
Large-scale efficacy comparison between Paclitaxel (PTX) versus colchicine (COL) of two maize genotypes. (**A**) V1 and (**B**) V2. SR (survival rate), DR (doubling rate), ADR (actual doubling rate), AER (anthers emergence rate), PFR (partial fertility rate), CFR (complete fertility rate), and TFR (total fertility rate). CK = control; this treatment is identical to all other treatments, but distilled deionized water (ddH_2_O) replaced PTX, colchicine, tween-80, and Di-methyl Sulfoxide (DMSO) solution. Different small letters on bars represent the significant differences within the treatments calculated using Tukey’s HSD test at *p* ≤ 0.05. Vertical bars on graphs indicate the standard error of the mean (*n* = 3).

**Figure 6 ijms-24-14659-f006:**
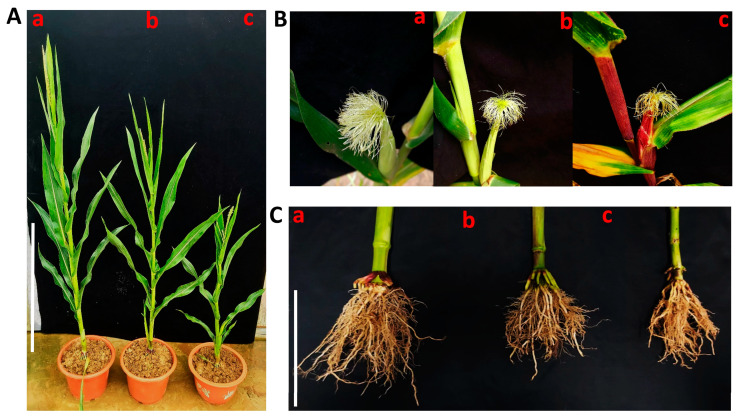
Comparative impact of paclitaxel (PTX) versus colchicine on the morphology of haploid maize plants. (**A**) Impact on plant and ear height; Scale = 85 cm. (**a**) CK (control), (**b**) PTX, (**c**) Colchicine. (**B**) Impact on silks number/ear (**a**) CK, (**b**) PTX, (**c**) Colchicine. (**C**) Root Growth (**a**) CK, (**b**) PTX, (**c**) colchicine; scale = 15 cm.

**Figure 7 ijms-24-14659-f007:**
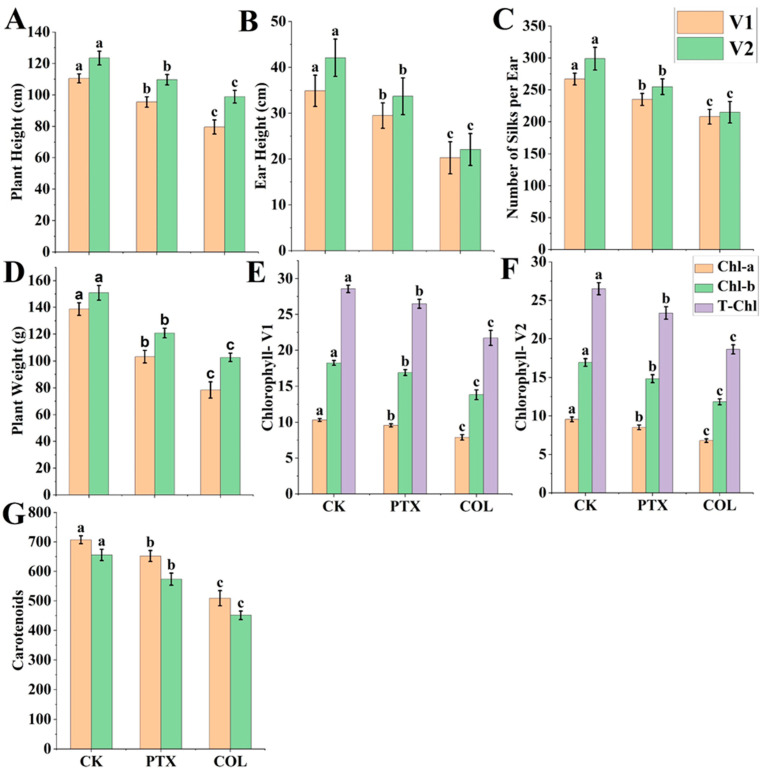
Comparative impact of paclitaxel (PTX) versus colchicine on morphology and physiology of haploid maize plants. (**A**) Plant height, (**B**) Ear height, (**C**) Number of silks/ears, (**D**) Plant weight, (**E**) Chlorophyll *a* (*Chl_a_*), Chlorophyll *b* (*Chl_b_*), and total chlorophyll (*Chl_T_*)—variety 1 (V1), (**F**) *Chl_a_*, *Chl_b_* and *Chl_T_*—variety 2 (V2), (**G**) Carotenoid contents. COL = colchicine; CK = control; this treatment is identical to all other treatments, but distilled deionized water (ddH_2_O) replaced PTX, colchicine, tween-80, and Di-methyl Sulfoxide (DMSO) solution. Different small letters on bars represent the significant differences within the treatments calculated using Tukey’s HSD test at *p* ≤ 0.05. Vertical bars on graphs indicate the standard error of the mean (*n* = 3 or *n* = 4).

**Figure 8 ijms-24-14659-f008:**
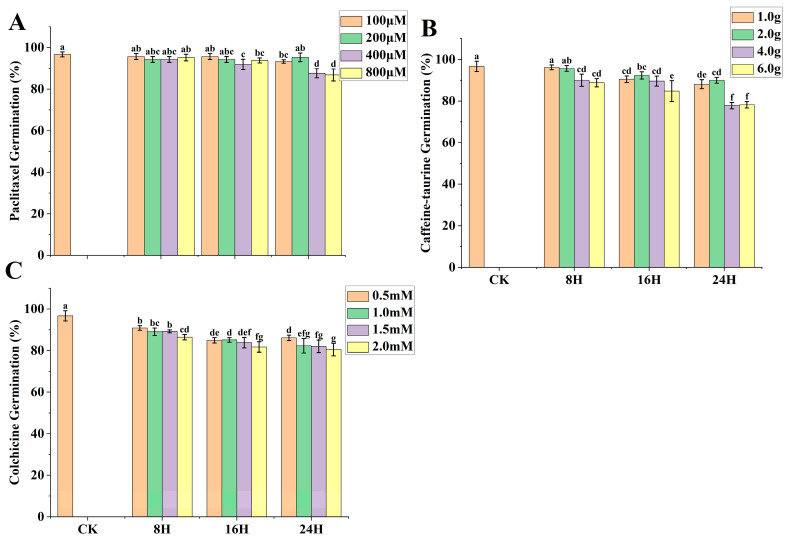
Comparative impact of paclitaxel (PTX), caffeine–taurine (CAF–T), and colchicine on seed germination with different concentrations and treatment time. (**A**) PTX impact on seed germination; (**B**) CAF–T impact on seed germination; (**C**) Colchicine impact on seed germination. CK = control; this treatment is identical to all other treatments, but distilled deionized water (ddH_2_O) replaced PTX, colchicine, tween-80, and di-methyl Sulfoxide (DMSO) solution. Different small letters on bars represent the significant differences within the treatments calculated using Tukey’s HSD test at *p* ≤ 0.05. Vertical bars on graphs indicate the standard error of the mean (*n* = 3 or *n* = 4).

**Figure 9 ijms-24-14659-f009:**
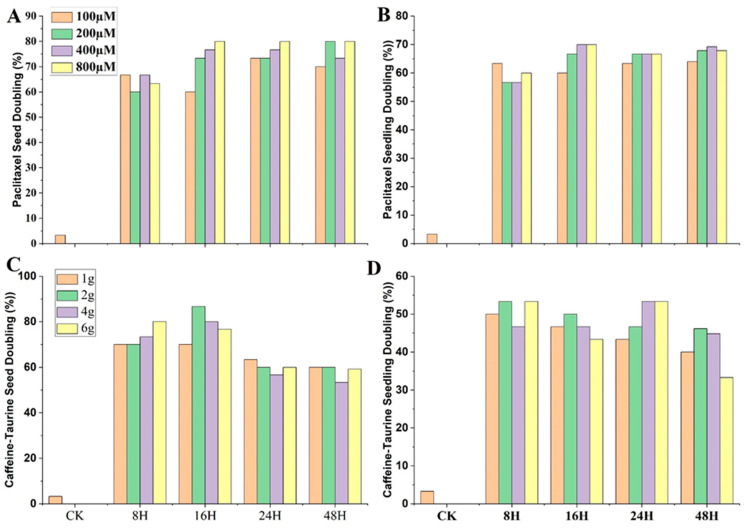
Confirmation of chromosome-doubling ratio (DR) under microscope induced by paclitaxel (PTX) and caffeine–taurine (CAF–T). (**A**) PTX-treated seeds; (**B**) PTX-treated seedlings (roots only); (**C**) CAF–T-treated seeds; (**D**) CAF–T-treated seedlings (roots only). CK = control; this treatment is identical to all other treatments, but distilled deionized water (ddH_2_O) replaced PTX, colchicine, tween-80, and Di-methyl Sulfoxide (DMSO) solution.

**Figure 10 ijms-24-14659-f010:**
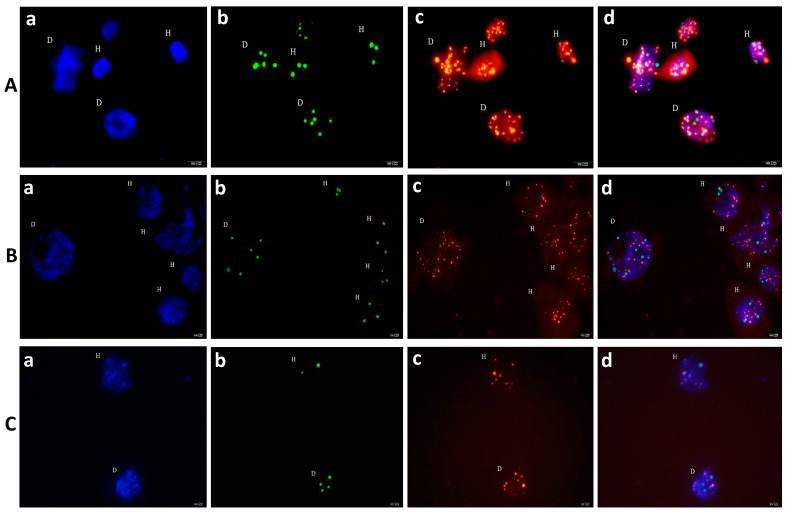
Fluorescence in situ hybridization (FISH) using knob-2 probe to detect chromosomal doubling signals induced by paclitaxel (PTX) and caffeine–taurine (CAF–T). (**A**) PTX-induced chromosomes doubled in the treated seed; (**B**) PTX-induced chromosomes doubled in treated seedling (roots only); (**C**) CAF–T-induced chromosomes doubled in the treated seed; (D) Diploid cells; (H) Haploid cells; (**a**) DAPI (4’,6-diamidino-2-phenylindole) (Vector) (blue); (**b**) Knob-2 (green); (**c**) Gypsy (red); (**d**) Merger of a, b, and c.

## Data Availability

The data presented in this study are already discussed in the main manuscript and Appendix A.

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
