# Peer review of "Paclitaxel and Caffeine–Taurine, New Colchicine Alternatives for Chromosomes Doubling in Maize Haploid Breeding"

_ijms, 2023, doi:10.3390/ijms241914659_

Round 1

Reviewer 1 Report

The study addresses the use of alternative agents than colchicine to develop maize viable diploids (2n) from sterile haploids (n). The authors assessed the use of two antimitotic and antimicrotubular agents, namely the Paclitaxel (PTX) and the Caffeine-Taurine (CAF-T) that are used in cancer therapy. Employing field experiments a set of critical morphological and physiological parameters were comparatively evaluated in maize diploids produced by PTX-, CAF-T-, and colchicine-treatments in response to various concentrations and treatment times.  Using fluorescence in situ-hybridization the chromosomal duplications were assessed in the produced diploids. The focus of the study is mainly a comparative assessment of antimitotic compounds without providing further insight of the molecular mechanisms involved in the process and/or the assessed parameters. Apart from quoting relevant references to the toxicity of the aforementioned compounds, the authors do not provide further data, thus limiting the scope of the study to a comparison of important characteristics and parameters for the development of diploids from sterile haploid plants. As such the study could be more suitable for a specialized agricultural journal.

Minor english editing is required.

Author Response

Response to Reviewer 1 Comments

Point 1: L14 The focus of the study is mainly a comparative assessment of antimitotic compounds without providing further insight of the molecular mechanisms involved in the process and/or the assessed parameters.

Response 1: Thanks for the suggestion! There are several antimitotic chemical agents being used for haploid chromosome doubling which are toxic and do not meet the expected results. Nowadays, the key objective of the studies is to find effective and less toxic chromosome-doubling alternatives. Thus, the primary objective of our study is to establish the effective application of paclitaxel (PTX) and caffeine-taurine (CAF-T) as new alternatives for haploid maize chromosome doubling. The molecular mechanisms involved in maize haploid chromosome doubling by PTX and CAF-T will be part of our future study. However, I have added the molecular mechanisms of PTX and CAF-T involved in cancer remedies.

Point 2: Apart from quoting relevant references to the toxicity of the aforementioned compounds, the authors do not provide further data, thus limiting the scope of the study to a comparison of important characteristics and parameters for the development of diploids from sterile haploid plants.

Response 2: Colchicine and antimitotic herbicides such as Pronamide, APM, trifluralin, and oryzalin have not been comprehensively studied regarding toxicity in maize crops so far. However, we have added the reported oral toxicity to medical rats into the draft.

Due to the unavailability of toxicity to maize, we compared our new chemicals with the most used colchicine as a control. We have measured the following parameters which may be affected after chemical treatment and have also never been studied together before in maize. (1) seed germination%, (2) survival rate, (3) anthers emergence, (4) partial fertility of tassel, (5) complete fertility of tassel, (6) pollen fertility, (7) plant height, (8) ear height, (9) number of silks/ear, (10) plant weight, (11) root growth, (12) chlorophyll a, (13) chlorophyll b, (14) total chlorophyll, (15) carotenoids, and, (16) seed setting.

Point 3: Minor English editing is required.

Response 3: We all authors paid very much attention to the English writing in the revision.

Reviewer 2 Report

This study considers novel alternative chemical treatments that induce genome doubling for use as part of double haploid breeding of maize. The technology is already an important component of maize breeding so this study reporting improvements in the technique will be of interest to plant breeders. The introductions starts by considering advantages of DH technology to plant breeding but perhaps a definition of the process itself at the start would be useful to readers who are unfamiliar with this technology. The remainder of the introduction provides well-referenced and up-to-date background to the topic that builds an argument for the need to explore new DH approaches in plant breeding. Clear study aims and approaches are described at the end of the introduction. Comprehensive and well organised results are presented comparing a range of treatments and treatment conditions. The photos are useful to represent the results but Figures 2 and 4 summarizing statistical results are a little crowded. I suggest to expand these figures to one full page each to better view the results. In comparison, Figures 3 and 5 look much better. The discussion develops the different contributions of this study to the topic, starting with comprehensive measurements of doubling rate efficency and toxity, but the main point of later sections is not always so clear. Addition of subheadings could make the logic of the organization more clear. Comparisons with the wider literature, including other plant species, are presented throughout for context. Methods are well organised and complete. The conclusions sum up the most relevant results of the study well making it a useful reference for plant breeders.

Specific comments
L14 Why specify "male" here?
L65-67 Mention the plant species tested in these studies.
L116-117 Specify the "another culture" referred to here.
L121-125 These lines return to discussing other DH treatments. I recommend to move this paragraph to before the paragraph describing potential new treatments for better logical flow of the paper.
L149 Replace "effected to" with "affected by"
L153 Add "different" after "non significant" here and elsewhere to make the test comparison more clear.
L333-334 I am confused by the two doubling rate ranges stated here (5-10 % and 60-80 %) How can both ranges apply?
L348-349 Same comment as above.
L510 Reference the haploid inducing treatment here.

Some minor edits suggested in my comments above.

Author Response

Response to Reviewer 2 Comments

Point 1: L14 Why specify "male" here?

Response 1: Replaced the word “male” with “tassel” in the draft. Haploid plants are male sterile due to a lack of homologous chromosomes leads to unequal chromosome distribution and eventually meiotic failure. As the cells get a set of chromosomes doubled in the tassel, the plant tassel may shed some pollen, so the tassel fertility (pollen-shedding) is considered an indication of the chromosome doubling of the haploid.

Point 2: L65-67 Mention the plant species tested in these studies.

Response 2:  Thanks, and we added the species “Maize”.

Point 3: L116-117 Specify the "another culture" referred to here.
Response 3: Yes, it has been added.

Point 4: L121-125 These lines return to discussing other DH treatments. I recommend moving this paragraph to before the paragraph describing potential new treatments for a better logical flow of the paper.

Response 4: Thanks, L121-L125 removed and shifted above

Point 5: L149 Replace "effected to" with "affected by"

Response 5: Thanks

Point 6: L153 Add "different" after "non-significant" here and elsewhere to make the test comparison more clear.

Response 6: Yes, non-significant difference.

Point 7: L333-334 I am confused by the two doubling rate ranges stated here (5-10 % and 60-80 %) How can both ranges apply?
Text: The M1 disclosed DR ranging from 60.0% to 80.0% across all treatments of PTX based on the frequency of 5-10% cells doubling.

Response 7: The M1 revealed DR ranging from 60% to 80% across all treatments of PTX based on chromosome doubling count under the microscope using root tip cells.

Point 8: L348-349 Same comment as above.

Text: The M1 revealed DR ranging from 53.3% to 86.7% across all treatments of CAF-T based on the frequency of 5-10% cells doubling.

Response 8: The M1 revealed DR ranging from 53.3% to 86.7% across all treatments of CAF-T based on chromosome doubling count under the microscope using root tip cells.

Point 9: L510 Reference the haploid-inducing treatment here.

Response 9: Sorry, the words “inducing treatment” are wrong. Revised into “Maize haploids seeds were produced at the College of Agriculture, Guizhou University”

Point 10: The introduction starts by considering the advantages of DH technology to plant breeding but perhaps a definition of the process itself at the start would be useful to readers who are unfamiliar with this technology.

Response 10: Yes, added some information at the beginning of “the introduction”

Point 11: The photos are useful to represent the results but Figures 2 and 4 summarizing statistical results are a little crowded. I suggest expanding these figures to one full page each to better view the results. In comparison, Figures 3 and 5 look much better.

Response 11: Expanded figures 2 & 4.

Point 12: The discussion develops the different contributions of this study to the topic, starting with comprehensive measurements of doubling rate efficiency and toxicity, but the main point of later sections is not always so clear. The addition of subheadings could make the logic of the organization more clear.
Response 12: Subheadings have been added as you suggested

Point 13: Comments on the Quality of English Language: Some minor edits were suggested in my comments above.

Response 13: Thanks!

Round 2

Reviewer 1 Report

The authors have addressed adequately all suggestions raised by the reviewers improving the manuscript.